# Design of Experiments to Compare the Mechanical Properties of Polylactic Acid Using Material Extrusion Three-Dimensional-Printing Thermal Parameters Based on a Cyber–Physical Production System

**DOI:** 10.3390/s23249833

**Published:** 2023-12-14

**Authors:** Miguel Castillo, Roberto Monroy, Rafiq Ahmad

**Affiliations:** Smart & Sustainable Manufacturing Systems Laboratory (SMART LAB), Department of Mechanical Engineering, University of Alberta, Edmonton, AB T6G 1H9, Canada; mecastil@ualberta.ca (M.C.); monroy@ualberta.ca (R.M.)

**Keywords:** FDM, cyber–physical production system, design of experiments, mechanical properties, thermal properties

## Abstract

The material extrusion 3D printing process known as fused deposition modeling (FDM) has recently gained relevance in the additive manufacturing industry for large-scale part production. However, improving the real-time monitoring of the process in terms of its mechanical properties remains important to extend the lifespan of numerous critical applications. To enhance the monitoring of mechanical properties during printing, it is necessary to understand the relationship between temperature profiles and ultimate tensile strength (UTS). This study uses a cyber–physical production system (CPPS) to analyze the impact of four key thermal parameters on the tensile properties of polylactic acid (PLA). Layer thickness, printing speed, and extrusion temperature are the most influential factors, while bed temperature has less impact. The Taguchi L-9 array and the full factorial design of experiments were implemented along with the deposited line’s local fused temperature profile analysis. Furthermore, correlations between temperature profiles with the bonding strength during layer adhesion and part solidification can be stated. The results showed that layer thickness is the most important factor, followed by printing speed and extrusion temperature, with very close influence between each other. The lowest impact is attributed to bed temperature. In the experiments, the UTS values varied from 46.38 MPa to 56.19 MPa. This represents an increase in the UTS of around 17% from the same material and printing design conditions but different temperature profiles. Additionally, it was possible to observe that the influence of the parameter variations was not linear in terms of the UTS value or temperature profiles. For example, the increase in the UTS at the 0.6 mm layer thickness was around four times greater than the increase at 0.4 mm. Finally, even when it was found that an increase in the layer temperature led to an increase in the value of the UTS, for some of the parameters, it could be observed that it was not the main factor that caused the UTS to increase. From the monitoring conditions analyzed, it was concluded that the material requires an optimal thermal transition between deposition, adhesion, and layer solidification in order to result in part components with good mechanical properties. A tracking or monitoring system, such as the one designed, can serve as a potential tool for reducing the anisotropy in part production in 3D printing systems.

## 1. Introduction

In additive manufacturing, in order to achieve mass production with controlled parameters and continuous quality outputs based on the mechanical properties required, process adjustments mostly depend on the interaction of printing parameters, material properties, and printer characteristics [1,2]. Therefore, tuning additional or individual parameters for the development of each product is time consuming and uses resources, producing waste either due to human error or machine variations [3]. Most of the time, these adjustments should be intuitive and user-friendly, but for a large fleet of printers with multiple part productions, this task seems impractical without utilizing any decision support system. For this purpose, the implementation of cyber–physical production systems (CPPSs) based on machine learning models (MLs), data science, and material properties knowledge-based models serve as the optimal solution. Suppose we consider the current state-of-the-art fused deposition modeling (FDM). It lacks automatized quality control tools and printing condition feedback systems, and it operates in an open loop due to the lack of conditions to adjust accordingly [4]. In this area, there have been studies applying novel image processing tools such as artificial intelligence combined with arrays of sensors to address and mitigate print failure, reducing waste, cost, and production time [4,5,6].

For 3D-printed parts, improving mechanical properties is a topic that has acquired great importance in recent years [7,8]. Several research papers have studied the correlation between printing parameters and mechanical properties, such as the ultimate tensile strength (UTS) [9,10,11], even after recycling cycles [12] or changes in the color of the filament [13]. The results of these studies have demonstrated the non-linearity of the process and the complex interaction between parameters; in most cases, this results in parts with anisotropic mechanical properties [11,14,15,16]. In the literature, there are publications that directly correlate the 3D printing parameters and influence of the final mechanical properties [10,11,17]. From the intelligent systems design standpoint there are numerous publications where several sensor array implementations have been utilized to enhance the autoregulation of the process parameters; nevertheless, there are few publications where the interconnection between parameters and process regulation systems simultaneously portray mechanical property relations. In 2022, Castillo et al. [7] published a review on smart manufacturing technologies applied to the material extrusion 3D printing process, where the need of a CPPS is an alternative solution to enhance and control the mechanical properties of 3D printing based on parameter monitoring for several materials. In this previous analysis, there is an extensive review of publications that correlate mechanical properties with 3D printing processing parameters. As for this study, the main experimental material foundations are presented in order to establish a knowledge base of the model materials for the purpose of building autoregulation process controls in the future.

Additionally, the correlation between the material temperature profiles during printing with the achieved mechanical properties is important, as it is a thermal process, and the correct temperature improves the adhesion between layers of the deposited material. In the process, adjacent fibers could be said to be deposited on top of one another. When one is molten, the other comes into contact within another in a semi-solid state; this creates an interface interaction between them called a layer bond [18]. To model the type of physics involved, consideration of a sequence of five steps is needed; initially, there is a surface rearrangement; then, a surface approach occurs, where intimate contact between the surfaces of each layer takes place; next, the wetting step occurs, where intermolecular forces are sufficient to start diffusing the polymer molecules with each other, creating a randomization of the particles and forming a bond. In each of the steps, the interaction between the rheological properties correlated to the temperature significantly determines the bond strength between each layer, and consequently, the final mechanical properties achieved [18,19]. The low temperatures prevent proper fusion between the layers; high temperatures promote fusion but, in turn, increase the fluidity of the material, so the part loses its shape, compromising the quality or inducing the degradation of the material [20,21,22]. Finding the optimal fused temperature profile will increase the mechanical properties without sacrificing quality. Some studies have evaluated the temperature profiles during printing using thermal cameras [18,23,24,25,26], while others have used thermocouples [5,27,28]. Both approaches have their pros and cons; to the knowledge of the authors, the average temperature of each deposited layer has been measured, but few studies have studied the material temperature at the extrusion point, as it is being deposited following the path marked by the G-code. This study seeks to contribute to that area.

The thermal behavior has been modeled in previous studies, showing promising results [18,27,29]. The deterministic approach could be the answer to improving the product’s mechanical properties if the temperature is the only or main variable to consider. Still, as previously explained, several parameters interact in a non-linear way in the process, producing anisotropies. This makes it difficult to find a deterministic solution. To address a multivariable problem with these characteristics, the authors consider that a viable option could be to explore statistical methods considering these interactions. Machine learning tools fit into this approach, but a material properties knowledge-based model must be developed through experiments to learn the behavior of the process parameter interactions and thus train a network to predict the final part of the mechanical results.

Therefore, CPPS are powerful tools for building a knowledge-based model and solving the problems of failed impressions and unpredictability in mechanical properties [30,31,32,33]. These systems seek to close the loop by taking data from the physical domain and processing it in the cyber domain. A 5C architecture helps to structure the system [34] and, in addition to ML tools and the Internet of Things (IoT) platforms, would facilitate the implementation and deployment of data, allowing real-time monitoring and remote control [35,36,37,38].

This study seeks to develop the material properties knowledge-based model through a design of experiments using Taguchi and full factorial analysis. The parameters were limited to four, so the number of experiments resulted in 52 specimens. The four parameters included two parameters with a direct influence on the temperature of the layers, such as polymer extrusion and bed temperatures. In addition, two parameters that impact the printing time were also included, such as printing speed and layer thickness, which affect the production cost for a part [39].

## 2. Cyber Physical Production System and Design of Experiments

### 2.1. CPPS and Data Acquisition

A large-scale 3D printer with a printing area of 610 mm × 1222 mm × 450 mm in the X, Y, and Z axes, respectively, was integrated in a CPPS. The 5C-level architecture proposed by [40] was implemented as a base manufacturing system for industry 4.0. Figure 1 presents the FDM system and the five levels. This study focuses on levels I, II, and III.

The first-level, smart connection belongs to the physical domain; the data are measured by an array of sensors deployed on the printer. In total, there are sixty different parameters being monitored using different sensors such as RGB and infrared cameras, accelerometers, thermocouples, and humidity, current, and voltage meters. Since the experiments carried out have the objective of studying the relationship between parameters, the deposited material temperature, and the ultimate tensile strength, the main sensor used was a 32 × 24-pixel small-size thermal sensor array (MLX90640). This was used to measure the temperature of the extruded material along with the remainder of the sensors in the system; its location can be seen in Figure 2. The camera is coupled to the printer head moving along with the nozzle; in this way, the pixels of interest are always fixed, and the local temperature at each polymer fused deposition line is traced.

The second data-to-information level belongs to the physical domain, which processes the data to transform it into meaningful information. A Python script was used to carry out all the processing algorithms. The data acquired were stored in a MySQL database. The script assigned a unique time stamp associated with the recorded data to differentiate each specimen. The G-code was also automatically modified to move the nozzle away at the end of each layer to be able to take a picture of all views of the part without the obstruction of the field of view; this condition will be important to interpret the data later. The reason for taking images is to be able to grow the database on this printer and to train artificial intelligence algorithms in the future. In this case, filters were not applied to the data since, at the third level, there was no need to do so at the time of viewing the data.

Finally, the cyber level is where the data are compared, either with the history of the same printer or with other printers in the production system; in this case, only one printer was used; therefore, the data were compared with the history of the same printer. For visualization of the process and the results, in addition to the dashboard, OriginPro 2023 software was used to produce the graphs. Then, the comparisons and relationships were carried out manually to develop the knowledge-based model; once it is developed, the processing will be automated, continuing with the fourth and fifth levels of architecture; these levels are not covered in this study. Figure 3 shows a graphic description of the designed cyber–physical production system.

### 2.2. Designs of Experiments

The printer is used with a super volcano hot end, and an extruder of 1.4 mm in diameter. The reason for this large nozzle size is that this printer was designed as a hybrid system where recycled material will be used as pellets for the infill through direct deposition and for better finishing of the filament system on the visible faces of the part (outer walls, and top and bottom layers). In order to guarantee a high throughput and slicing compatibility between both deposition systems, a nozzle size of a similar diameter to the pellet nozzle was used; more information can be found in [41].

The material selected was PLA-grade standard filament manufactured by AMZ3D, with a 1.75 mm diameter, standard extrusion temperature from 210 to 230 °C, and bed temperature of 50–70 °C. To maintain consistency, all of the tests were carried out using the same color. To guarantee that no moisture was present in the filament, a new spool was used, which was stored in a closed bag of white silica gel to reduce its exposure to the environment; it is also important to highlight that according to the sensor in the CPPS, the humidity was never above 16%. As can be seen in Figure 4, the printer does not have a closed chamber; inevitably, the parts were exposed to changes in the environmental temperature; for example, when the doors of the laboratory were opened, the temperature might have changed. The limitation of not having a closed chamber hinders the use of other types of filaments, such as ABS or HIPS, since these materials are prone to warping due to changes in temperature.

One of the goals of this study is to understand the thermal interactions during printing and relate them to the mechanical strength of the part produced. It has been shown in previous studies that the bonding strength between layers is a key factor in improving the mechanical properties of an object [23,24,25,26,27,42,43]. Therefore, the factors selected for analysis are extrusion temperature, bed temperature, printing speed, and layer thickness, since they are related to the extruded material temperature. Two methods were implemented, starting with a Taguchi analysis to rank the influence of the parameters in the ultimate tensile strength; this method makes it possible to analyze the effect of multiple parameters with few tests with the help of orthogonal arrays [44,45]. This analysis ranks the parameters according to their influence, detecting the main variations in the average statistical results, facilitating fewer sample analyses with the same scope of observation. Finally, only the two most relevant parameters will be selected to be included in a full factorial experiment to test all the possible combinations of parameters and levels.

For the Taguchi analysis, the size of the array depends on the number of parameters and the levels of each of them; therefore, the greater the number of parameters and levels, the greater the array. The selected levels for each parameter are shown in Table 1, as well as the values of the remaining printing fixed parameters. Starting with the polymer extrusion temperature, the manufacturer recommends a temperature of 190 to 220 °C for this filament, so the range was divided into three levels evenly. Then, for the printing speed, considering the goal of the designed printer, which is to achieve high throughput, the range was selected as being 30 to 50 mm/s. For the bed temperature, the commonly used values are around 70 °C, and a range from 60 to 100 °C was selected. Finally, for the layer thickness, since Ultimaker Cura 5.3.1 was used to generate the G-code for the printer, their default profiles range from extra fine at 0.06 mm up to extra course at 0.6 mm; thus, the three highest, evenly divided profiles of 0.2, 0.4, and 0.6 mm were selected.

To describe the Taguchi analysis, this statistical method relies on the signal-to-noise ratio, as described in the equation, whereas *y_i_* and *s_i_* are the average and variance of all the data values of the *i*th experiment. Ideally, the SN ratio should be a high value for good product quality because it means that the noise presented in the system is comparatively lower than the signal itself; in other words, the parameter being studied has little effect on the system performance. The advantage of reducing experiments on this analysis comes with an important drawback since it does not consider all the possible combinations the results are relative, so the influence ranking might not be exact. To mitigate this issue, a full factorial analysis is carried out later with the first two ranked parameters.
$${SN}_{i}=10log\frac{y_{i}^{2}}{s_{i}^{2}}$$

Following the Taguchi analysis procedure [45], for four different parameters with three levels, the L-9 array was carried out, where nine different combinations are printed and tested. To consider the possible variability, three specimens for each combination were printed and the average of the test was taken to detect possible abnormalities in the results. A total of 27 specimens were printed, as can be seen in Table 2.

#### 2.2.1. Three-Dimensional Printing of SPECIMENS

The printer used was a large-format machine with a hybrid extrusion system; the infill pattern used was grid since it showed better performance on the tensile test because the force applied goes along with the axis printed [11,50]. The build orientation was flat as it provides a wider surface of support on the subtracted layers [39,51,52]. The infill density was kept as the default value of 20% of the slicer software, and the layer width was 0.8 mm since a nozzle of 1.4 mm was found to achieve good printing results according to the trials. It was also noted that no build plate adhesion was needed for these conditions.

#### 2.2.2. Tensile Testing of Specimens

The standard followed was the ASTM D638 (Type I); the shape of the printed part, also known as the dog bone specimen, can be observed in Figure 5. Each specimen was printed one at a time; pictures and video of X, Y, and Z views, thermal camera data, and the rest of the CPPS sensors (RGB camera, thermocouples, accelerometers, voltmeters, and so on) were recorded with a unique time stamp for each specimen for its further analysis.

The equipment used to conduct the destructive test can be seen in Figure 5; an Instron 5966 with a load cell of 10 kN and a gripper with a maximum load of 5 kN were used. The test was performed at a displacement rate of 1 mm/min. To measure the axial strain, an Epsilon ONE camera was used, which followed two marks on each side of the specimen, as can be seen in bottom right corner of Figure 6.

## 3. Experimental Results

### 3.1. Taguchi Analysis

#### 3.1.1. Mechanical Test Results

After printing the 27 specimens, the destructive test was carried out, and the stress–strain curves were plotted; some of them can be seen in Figure 7. Then, the ultimate tensile strength was obtained for each test, which is the maximum stress in the stress–strain curve.

Table 3 shows the UTS values for all 27 specimens ranked according to the average value from the three tests. It is important to note that since the layer height was changed for three different levels and the layer height was 3 mm according to the standard, the number of layers that provided exactly the expected height for the layer thicknesses of 0.2 and 0.6 mm were 15 and 5, respectively. For the 0.4 mm layer thickness, the slicer software used 8 layers for a final height of 3.2 mm, which is something that usually happens when the layer thickness is fixed. Then, the slicer software rounded the number of layers up or down based on the value that was closest to that which was expected. In this case, it was 8 layers, adding 0.2 mm of extra height. To compensate for this difference and be able to rank the parameters, 6% of the extra height was subtracted from the UTS value, which can be seen in the last column of Table 4, in relation to the average UTS value compensated (Avg. Comp). In general, it can be seen that when ranking from the highest UTS to the lowest, for example, the first positions are occupied by the specimens with the highest layer thickness (LT). This is not a pattern that is strictly followed since there are three other parameters that affect the results. For the extrusion temperature (ET), the printing speed (PS), and the bed temperature (BT), there is not such an obvious pattern in terms of influence; thus, Taguchi’s analysis clarifies the contribution of each one. From the highest UTS of 56.19 MPa to the lowest at 46.38 MPa, there is a variation of 9.81 MPa. This represents 17.45% and is an example of the influence of the different levels in each parameter. Table 4 presents the failure strain list for the specimens.

#### 3.1.2. Analysis of Results—Taguchi

For the Taguchi L-9 array, the results are listed in Table 5. The goal is to compare the impact of parameters related to the thermal interaction of the process with the ultimate tensile strength. The first step is to calculate the mean and variance of each experiment and then the signal (average value) to noise (variance) ratio.

Subsequently, to rank the parameters, the delta value needs to be calculated. This is defined as the maximum noise-to-signal ratio minus the minimum; the results are given in Table 6. The layer thickness occupies the first position with a delta of 7.88; its ranking is almost twice that of the others, which supports the observations of the previous section; it stands out as the parameter with the most influence. In the second and third position are printing speed and extrusion temperature, respectively, with almost the same value of around 4; due to the closeness of the values, this might be one of the cases where Taguchi is not accurate enough to guarantee an exact position; if all the possible combinations are considered, then the second and third position might change. Finally, in fourth place is bed temperature with a delta of 0.88; with this value, it can be concluded that its influence on the UTS is small. Considering these results, the first two parameters were used for a full factorial analysis.
$$∆={SN}_{i}\max-{SN}_{i}\;min$$

#### 3.1.3. Analysis of Results—Thermal Interaction Behavior

The parameters used in this analysis will be reviewed in detail in this section to understand the thermal interaction during printing.


**Extrusion temperature analysis**


The CPPS was used to store the data related to each specimen; they were printed one at a time. The local fused temperature of the extruded material was plotted; some profiles can be seen in Figure 8, where specimens 2 and 3 are part of experiments 1, 8, and 9 in the first and second graphs, respectively, and specimens 25 and 27 are part of the fifth and sixth graphs, respectively. In general, it can be observed that there was not much of a difference between the different printed parts in the same experiment, which is supported by the low variance in the UTS of specimens with the same combination. It is also noticeable that the local fused temperature of the extruded material is lower than the heater block temperature; this difference is tabulated in later sections. Finally, for every layer change, there is a sudden drop in the temperature due to the nozzle moving away from the part to take the pictures without visual obstructions.

In a more detailed observation, the temperature varies widely at around almost 10 °C of difference, and considering the points where the nozzle moved away to take the images, this difference is even greater. From the layer line, it can be noticed that even layers reach a higher temperature than odd layers; this phenomenon can be explained by studying the thermal camera videos that focus the nozzle. The path follows a grid pattern of 0 and 90°, where the odd layers are at 0° (movement on Y), with a longer path along the axis of the dog bone specimen, giving enough time for the material to cool down. This is in contrast to the 90° (movement on X) layers, where the path is much shorter, so not only does the layer not cool down, but it also has more influence from the adjacent layers. Figure 9 shows this pattern along with the registered movements in X and Y; a clear relationship can be seen. These findings indicate the influence on the process of the tool path, from a thermal point of view.

In general, the local fused temperature presented notable variations. The heat transfer between the heat block and the PLA is affected by the movement in X and Y; later, this will also be seen to be true for the other parameters used. In a dynamic process, it is difficult to guarantee an optimal and constant heat transfer that equalizes the temperature of the material to the temperature of the block; therefore, the local fused extruded material is at a lower temperature.


**Printing speed analysis**


For the three levels of speed, the average temperature per layer and the difference from the requested temperature were calculated; the results can be seen in Figure 10, where the faster the nozzle moves, the cooler the material is. The temperature falls because the heater block has less time to melt the filament; thus, the difference in the requested temperature increases. This result is interesting because this parameter ranked second in Taguchi’s analysis, but comparing the temperature, there is only one degree of difference from 30 to 50 mm/s, which represents 2.69%; its influence on the temperature is less than the influence of the bed temperature, which leads one to think that the improvement in the UTS due to printing speed is not only related to temperature.


**Bed temperature analysis**


For the bed temperature, the data acquired by the CPPS is shown in Figure 11. On the right side of the graph, the bar shows the average temperature difference in the deposited material for each level of the temperature of the bed; it can be seen that as the bed temperature increases, the difference is reduced, with a 6.78 °C difference from 60 °C to 100 °C, down to a 5.6% difference. The right side of Figure 11 presents a comparison against the ideal (requested) temperature. The bed temperature influences the temperature of the layers. However, considering the result of the Taguchi analysis, since this parameter has the lowest influence, the result is 5.6%, compared to 2.69% for the printing speed. The bed temperature does not have a large influence on the UTS.


**Layer thickness analysis**


The results are presented in Figure 12; the left side shows a difference of 4 °C from 0.6 to 0.2 mm, which is 10.6%. This indicates that the greater the layer thickness, the smaller the difference. The right side of Figure 12 shows how the parameter caused the temperature to be much closer to the desired temperature (in contrast to the previous result). Taguchi’s analysis indicated that this has the greatest influence on the UTS, and we can see how temperature might be related, since 10.6% surpasses the thermal variation in all the previous parameters.

### 3.2. Full Factorial Analysis

#### 3.2.1. Mechanical Test Results

Following the results of the Taguchi analysis, the two parameters with the greatest influence on the UTS were selected, in this case, layer thickness and printing speed. Meanwhile, extrusion temperature and bed temperature were kept constant at their medium levels of 205 °C and 80 °C, respectively. All possible combinations were determined for the three levels of thickness and speed, printing three specimens for each case and thus being able to later calculate the averages; the results can be seen in Figure 13 for some of the specimens.

The values obtained are tabulated in Table 7, using the ranking of the last column compensating for the extra 0.2 mm that specimens with a layer thickness of 0.4 mm have, due to the rounding approximation of the slicing software. On the one hand, as the thickness increases, the UTS average increases. On the other hand, as the printing speed increases, the UTS value gradually decreases. The lowest value of UTS occurred with the most unfavorable combination, that is, a 0.2 mm layer thickness and 50 mm/s printing speed, reaching a UTS of 50.72 MPa; on the contrary, the highest value was reached with 0.6 mm layer thickness and 30 mm/s printing speed with a UTS of 54.86 MPa, a difference of around 4 MPa, representing a 7.5% improvement.

#### 3.2.2. Analysis of Results—Full Factorial

Figure 14 plots the results of Table 7; on the left side is the UTS average where the curve for the layer thickness of 0.4 mm exceeds 0.6 mm due to the extra 0.2 mm thickness; on the right side, the adjusted values represent the 0.6, 0.4, and 0.2 mm layer thicknesses in descending order of the UTS. For the three-layer thicknesses, the drop in the UTS value is not linear since the slope from 30 to 40 mm/s is less significant compared to the slope from 40 to 50 mm/s. In terms of the changes in UTS, they are not linear either, since the increase from 0.4 to 0.6 mm is at least three times greater than the increase from 0.2 to 0.4 mm. The three curves show a defined trend, corroborating the validity of the results. In the next section, we will seek to relate this behavior to the thermal interaction that occurred during printing.

#### 3.2.3. Analysis of Results—Thermal Interactions Behavior

Since the parameters were reduced using the results of the Taguchi analysis and all combinations were tested for layer thickness and printing speed, the thermal interaction can be easily studied below.


**Layer thickness analysis**


Using the data collected with the CPPS, the average local fused temperature of the extruded material and the difference compared with the requested temperature was calculated for each layer thickness; this can be seen in Figure 15. Similar to the conclusions found with the Taguchi analysis, as the layer thickness increased, the average temperature also increased, reducing the difference between the actual and the requested temperature. From layer thicknesses of 0.2 to 0.6 mm, there is 4.49 °C of difference or 2.81%. This relationship could influence the increase in the UTS value.


**Printing speed analysis**


Regarding the analysis of the printing speed, the right side of Figure 16 shows that as the speed increased, the temperature of the material decreased, and the difference between the actual and the desired temperature increased. Next, the average temperature values of the fastest and slowest layers of each of the printed specimens were taken in order to compare the results; these are shown on the right side of Figure 16. The fastest layers reached an average of 161.43 °C and 163.68 °C for the slowest layers, representing a difference of 2.25 °C, which is 1.39%. It can be concluded that the faster the filament passes through the heater block, the lower the temperature of the material in the nozzle will be because there is less time for heat transference and possible material thermal conductivity properties.

## 4. Discussions

The important aspects of the experimental findings are discussed in this section. Initially, based on the literature, the parameters used to analyze the thermal interactions between mechanical properties and process parameters were selected because they should have a direct influence on the local fused material temperature profiles of each layer; this hypothesis was found to be true, even when some had less influence. Even though the bonding strength between layers is an important parameter in the UTS of the printed part, for all the cases, an increase in the temperature was translated to an increase in the UTS value. This influence cannot be related only to the thermal increase in the temperature, since, for example, bed temperature had a lesser influence according to the Taguchi analysis. However, it was associated with 5.6% of the difference in the temperature in comparison to the printing speed with the second position, with a 2.69% temperature difference. This study also showed how the temperature of the material fluctuates throughout the process, being affected by the tool path in a complex interaction between different variables, which contributes to the product anisotropy. For the three specimens printed, the UTS, variances, and temperature profiles were close to each other, showing that parts printed with the same parameter combination do not show large variations; this was a pattern followed by the 54 specimens printed for both analyses. The data acquired by the CPPS showed logical and consistent trends. No problems were detected; therefore, during the development of these experiments, it was a reliable tool for analysis. All the data of the 54 impressions were stored in the database, allowing us to study and relate more variables, and possibly generate more results and conclusions. Although, during the course of the experiments, the ambient temperature could have changed when the laboratory door was opened since there was no printer chamber, it was difficult to see these disturbances in the temperature of the extruded material. This could indicate that warping affects the layers that are already deposited (temperatures that were not monitored) to a greater extent than the layer that is being extruded (temperature recorded). This is due to the fact that residual stresses can be generated during the solidification process. If the temperature gradient is higher, the compression forces during layer solidification can be greater than the tensile forces during melting. This can result in the part containing residual stresses that can affect the mechanical test results.

The 14.8% variation in the experiments for Taguchi and the 7.5% variation in the full factorial for the UTS value showed the opportunities to control and increase the mechanical properties of a product, with fixed design parameters. Therefore, there must be an optimal state for all of the parameters that allows the achievement of the highest mechanical performance without sacrificing, for example, the quality of the part or increasing the printing time, which results in higher costs. Furthermore, to extend discussion of the previous results, the following must be considered:From the initial thermal measurements, it was considered that the influence of the extrusion temperature and bed temperature would have a significant impact on the final UTS values due to the direct effects of temperature on material degradation properties. However, this being a thermal process involving bonding, and surface and diffusion interactions, the more prominent factors based on the Taguchi analysis were the layer thickness and printing speed parameters.The fact that those parameters significantly affected the final mechanical properties with the same design factors means that it is important to model and monitor the thermal conduction and convection phenomena during the process. This is something that would have been difficult to observe when experimenting with individual parameters.The temperature gradient from the extrusion temperature in the block to the deposited temperature was found to be significantly reduced. Furthermore, if the printing speed was increased, the bonding strength was decreased, in addition to the final mechanical properties. This leads to the fact that the material thermal diffusivity plays a significant role during the process, and the material requires an optimal transition from a liquid to a solid state during bonding to achieve the optimal mechanical properties.The existence of a significant temperature difference between longer printing paths where more thermal diffusivity allows rapid solidification and less bonding interaction was observed, compared to shorter paths with a higher local fused temperature. This corroborates the idea that the design toolpath can induce anisotropy in the process and affect thermal interaction between the layers.From the experimental material model and the employment of the CPP system to track thermal variations, it was found that this tool can have the potential to monitor the process in situ and the ability to predict the quality characteristics of printed parts.

## 5. Conclusions

This study used the design of experiments to evaluate the influence of four printing parameters and the ultimate tensile strength of specimens that followed the standard ASTM D638 (Type I). The parameters were chosen based on their direct influence on the thermal behavior of the process and the printing time. Using a CPPS, the thermal profile was analyzed with the objective of relating the UTS to the temperature based on its importance in relation to the bonding strength to achieve good adhesion between the layers, improving the UTS. Two methods were used: a general analysis, which is based on Taguchi’s orthogonal arrays that allowed evaluation of the influence with a reduced number of experiments, and a full factorial analysis that is more detailed in the two most important aspects according to Taguchi. The results showed that the order of the parameters in decreasing order of importance is layer thickness, printing speed, printing temperature, and bed temperature. Then, it was possible to graph the influence of the layer thickness and the printing speed on the UTS value. It was observed that the slope of the curves was not linear throughout the three levels studied, nor was it linear in terms of the contribution to the value of UTS. In general, the 34 printed parts showed that an increase in the average temperature during printing improved the UTS value. This supports the hypothesis of the importance of temperature in terms of good layer adhesion. There is a marked difference between the temperature of the heater block and the extruded material, for example, the highest temperature set for the heater block was 220 °C, but the material only reached 190 °C according to the thermal camera. This represented a difference of 30 °C in temperature. Intuitively, one might think that the temperature of the material can be increased without reaching its degradation value, thus improving the UTS; however, in practice, there comes a point where the material becomes very soft and flimsy, and the shape is lost in the deposited walls, preventing a quality piece from being printed. The influence of the four parameters studied in terms of the UTS cannot be demonstrated overall. The same is true for the influence of temperature alone. This is supported because even though an increase improved the UTS, the bed temperature, for example, provided a greater increasing delta compared to printing speed. Regardless, it finished in last position in terms of its influence on the UTS.

## Figures and Tables

**Figure 1 sensors-23-09833-f001:**
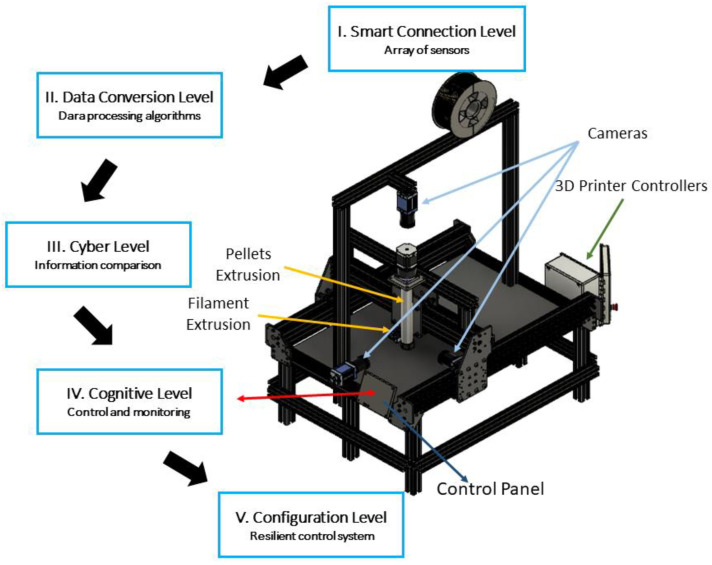
CPPS for a large-scale 3D printer with a hybrid filament and pellet system (adapted from [34]).

**Figure 2 sensors-23-09833-f002:**
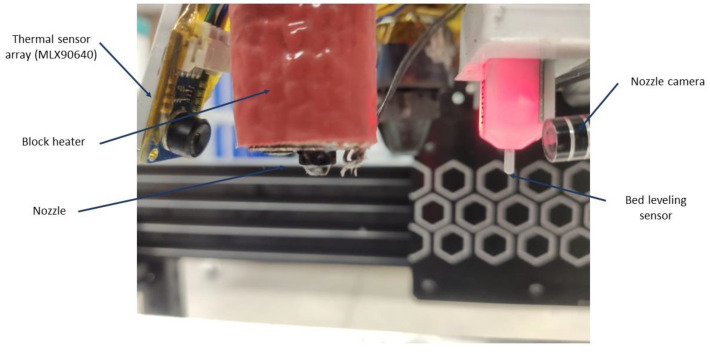
Thermal camera focusing the nozzle.

**Figure 3 sensors-23-09833-f003:**
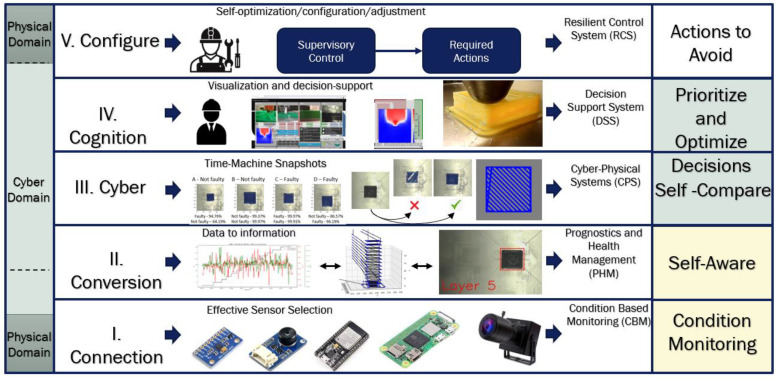
Graphic description of the designed CPPS.

**Figure 4 sensors-23-09833-f004:**
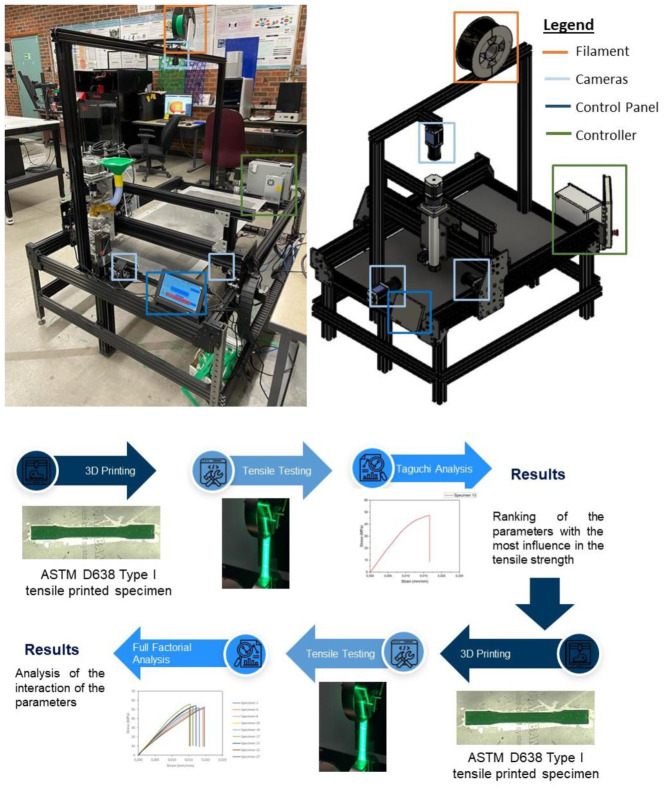
Large-format 3D printer, composed of a hybrid system of filament and recycled pellets in a high-throughput configuration to print large parts; on the left is the printer in the physical domain, on the right is the CAD design in the cyber domain, and at the bottom are the experiments to be performed.

**Figure 5 sensors-23-09833-f005:**
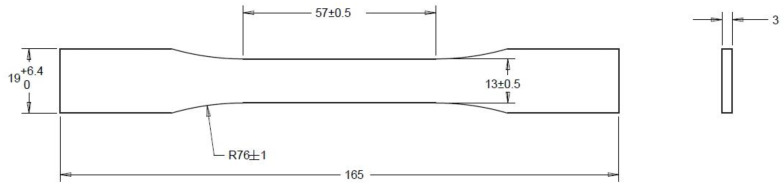
Dog bone specimen (ASTM-D638).

**Figure 6 sensors-23-09833-f006:**
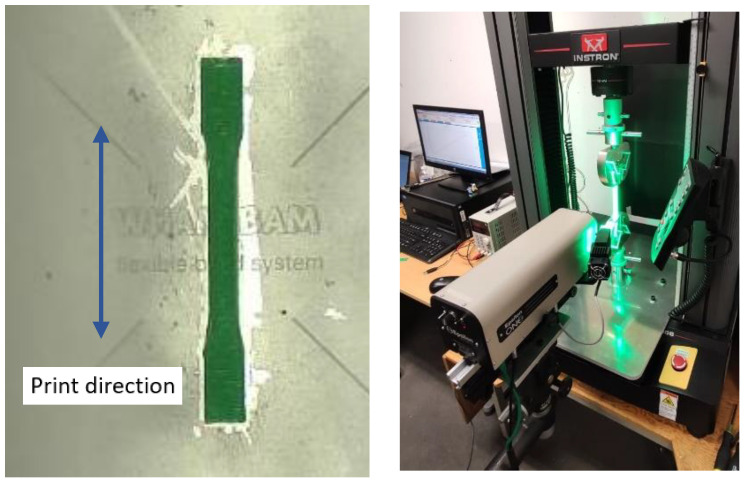
Specimen and equipment used for tensile testing.

**Figure 7 sensors-23-09833-f007:**
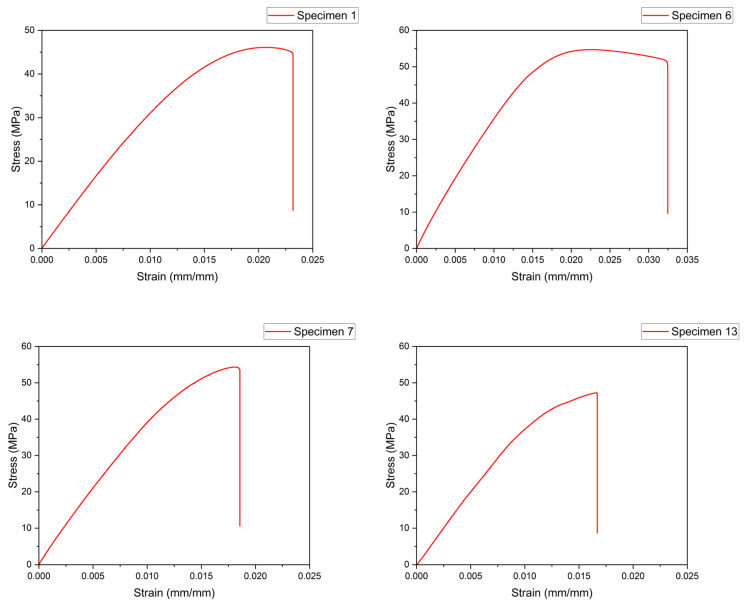
Stress–strain analysis for Taguchi specimens.

**Figure 8 sensors-23-09833-f008:**
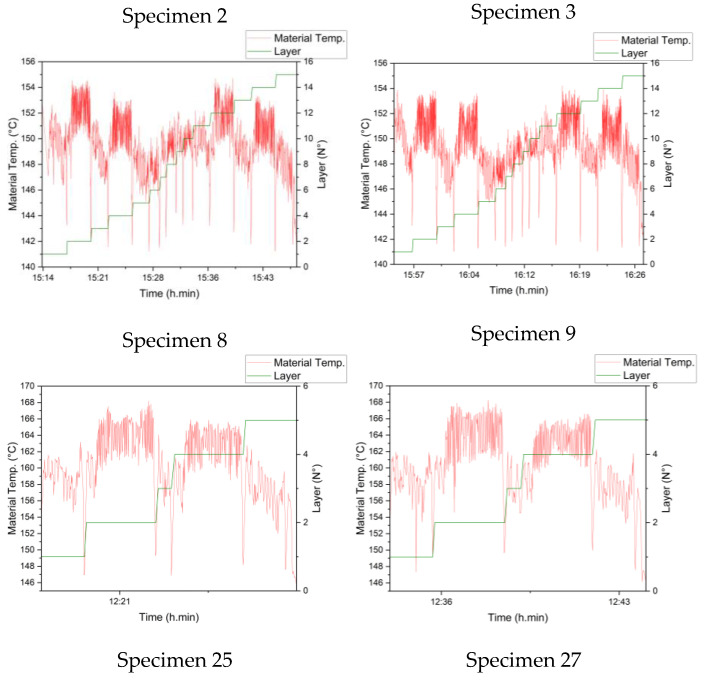
Local fused material temperature vs. layer.

**Figure 9 sensors-23-09833-f009:**
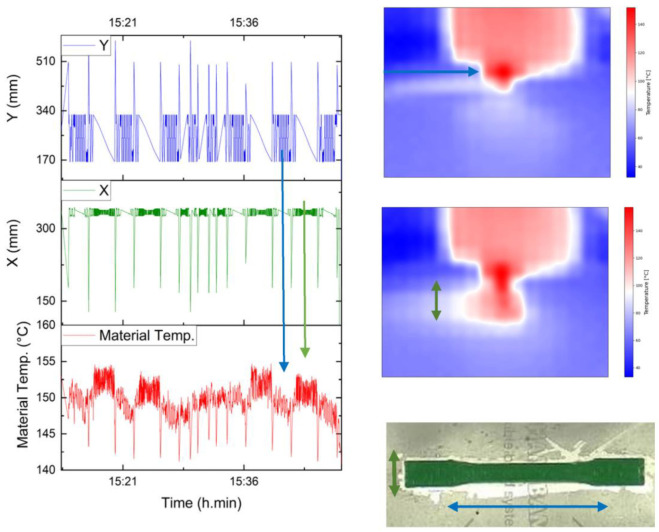
Even and odds layers vs. movement in X and Y.

**Figure 10 sensors-23-09833-f010:**
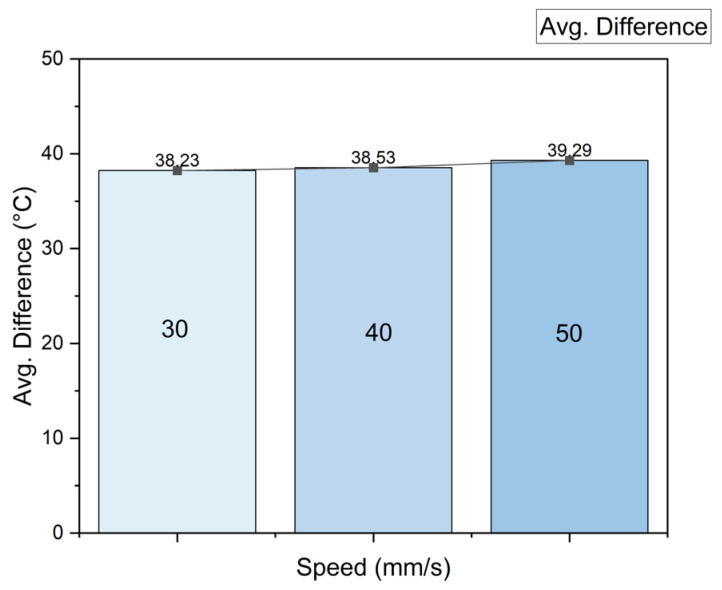
Material temperature vs. printing speed.

**Figure 11 sensors-23-09833-f011:**
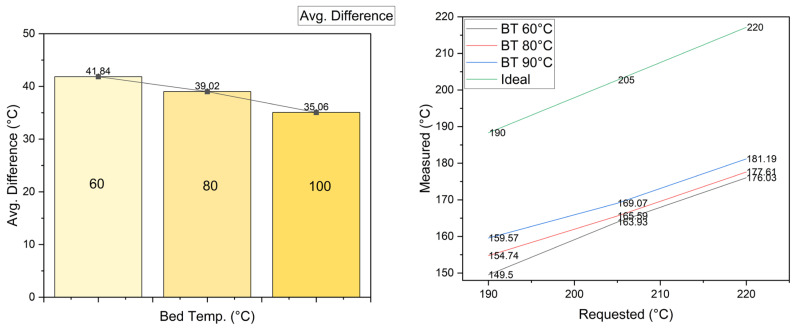
Material temperature vs. bed temperature.

**Figure 12 sensors-23-09833-f012:**
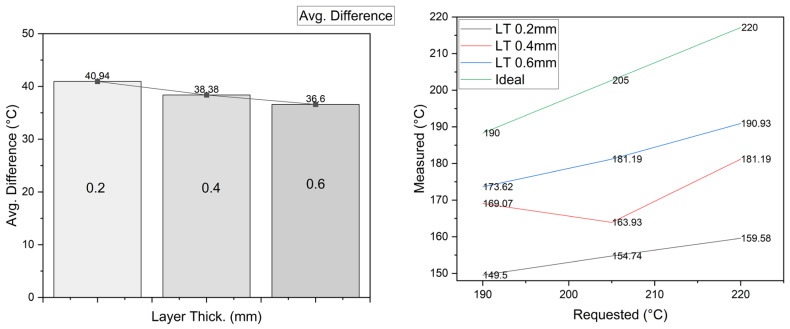
Material temperature vs. layer thickness.

**Figure 13 sensors-23-09833-f013:**
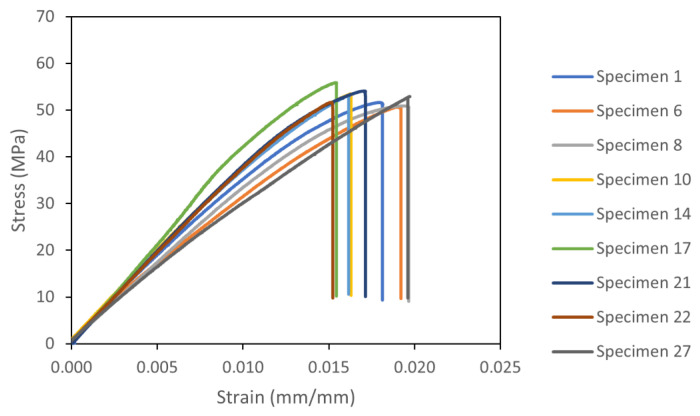
Stress–strain analysis for full factorial specimens.

**Figure 14 sensors-23-09833-f014:**
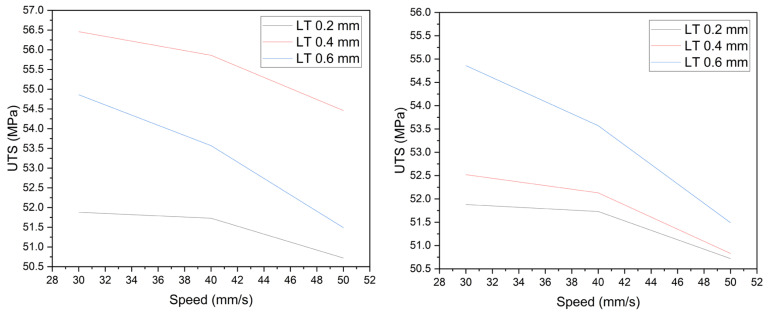
Full factorial results of layer thickness and printing speed.

**Figure 15 sensors-23-09833-f015:**
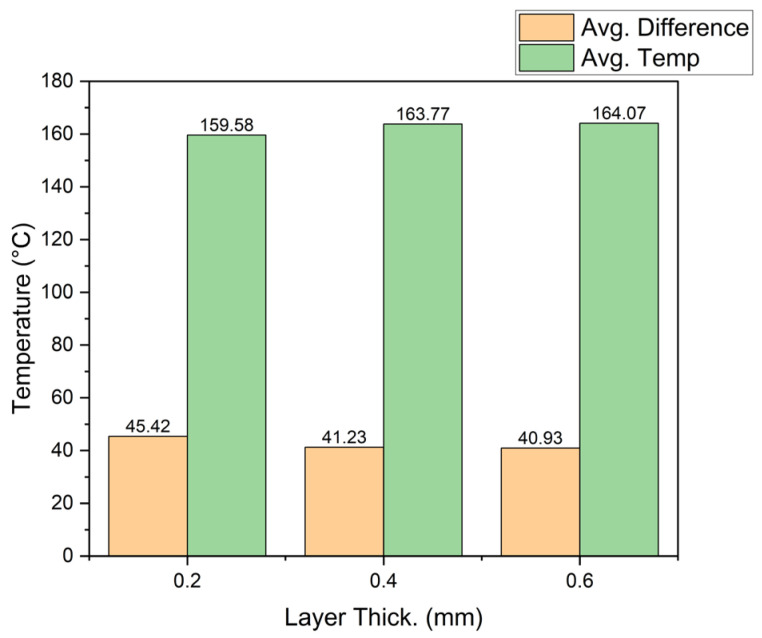
Layer thickness vs. material temperature.

**Figure 16 sensors-23-09833-f016:**
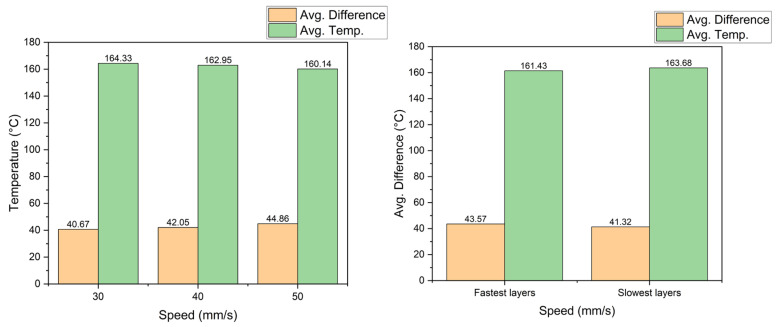
Printing speed vs. material temperature.

**Table 1 sensors-23-09833-t001:** Parameters selected for printing.

Parameter	Value	Source
Extrusion temperature (°C)	190, 205, 220	Suggested by the filament manufacturer
Printing speed (mm/s)	30, 40, 50	Limited by printer capabilities [46,47]
Bed temperature (°C)	60, 80, 100	[48,49]
Layer Thickness (mm)	0.2, 0.4, 0.6	Following the suggestion of the slicer software
Infill pattern	Grid	[11,12,50]
Build orientation	Flat	[12,51]
Infill density	20%	Default value of Ultimaker Cura
Layer width	0.8 mm	Based on printing trials for a 1.4 mm nozzle
Build plate adhesion	None	For PLA, this is not needed

**Table 2 sensors-23-09833-t002:** Design of Experiments for the Taguchi Analysis.

Experiment N°	Specimen N°	Extrusion Temp.	Printing Speed	Bed Temp.	Layer Thick.
1	1, 2, 3	190	30	60	0.2
2	4, 5, 6	190	40	80	0.4
3	7, 8, 9	190	50	100	0.6
4	1, 2, 3	205	30	80	0.6
5	4, 5, 6	205	40	100	0.2
6	7, 8, 9	205	50	60	0.4
7	10, 11, 12	220	30	100	0.4
8	13, 14, 15	220	40	60	0.6
9	16, 17, 18	220	50	80	0.2

**Table 3 sensors-23-09833-t003:** UTS results for Taguchi specimens.

Experiment N°	Specimen N°	ET	PS	BT	LT	Sample 1	Sample 2	Sample 3	Avg.	Avg. Comp
		°C	mm/s	°C	mm	Ultimate Tensile Strength MPa			MPa	MPa
8	22, 23, 24	220	40	60	0.6	50.48	58.02	54.94	54.48	54.48
3	7, 8, 9	190	50	100	0.6	54.33	55.07	51.01	53.47	53.47
7	19, 20, 21	220	30	100	0.4	55.46	57.32	55.79	56.19	52.44
4	10, 11, 12	205	30	80	0.6	52.23	49.26	54.08	51.86	51.86
2	4, 5, 6	190	40	80	0.4	54.47	56.40	54.70	55.19	51.51
9	25, 26, 27	220	50	80	0.2	50.61	52.75	50.91	51.42	51.42
5	13, 14, 15	205	40	100	0.2	47.23	47.20	50.42	48.28	48.28
6	16, 17, 18	205	50	60	0.4	49.05	50.53	51.64	50.41	47.05
1	1, 2, 3	190	30	60	0.2	46.10	47.18	45.87	46.38	46.38

**Table 4 sensors-23-09833-t004:** Failure strain list.

		A	B	C	D	Failure Strain mm/mm		
Experiment N°	Specimen N°	ET	PS	BT	LT	Sample 1 (T1)	Sample 2 (T2)	Sample 3 (T3)
1	1, 2, 3	1	1	1	1	0.0231	0.0253	0.0235
2	4, 5, 6	1	2	2	2	0.0336	0.0320	0.0324
3	7, 8, 9	1	3	3	3	0.0187	0.0167	0.0172
4	10, 11, 12	2	1	2	3	0.0130	0.0147	0.0138
5	13, 14, 15	2	2	3	1	0.0178	0.0182	0.0164
6	16, 17, 18	2	3	1	2	0.0181	0.0196	0.0172
7	19, 20, 21	3	1	3	2	0.0172	0.0184	0.0166
8	22, 23, 24	3	2	1	3	0.0192	0.0214	0.0218
9	25, 26, 27	3	3	2	1	0.0239	0.0214	0.0227

**Table 5 sensors-23-09833-t005:** Results of the tensile test and Taguchi analysis.

		A	B	C	D	Ultimate Tensile Strength MPa			Taguchi Analysis		
Experiment N°	Specimen N°	ET	PS	BT	LT	Sample 1 (T1)	Sample 2 (T2)	Sample 3 (T3)	Avg. (y_i_)	Variance (S_i_^2^)	Signal-to-Noise ratio (SN_i_)
1	1, 2, 3	1	1	1	1	46.10	47.18	45.87	46.38	0.49	36.42
2	4, 5, 6	1	2	2	2	54.47	56.40	54.70	55.19	1.11	34.39
3	7, 8, 9	1	3	3	3	54.33	55.07	51.01	53.47	4.67	27.87
4	10, 11, 12	2	1	2	3	52.23	49.26	54.08	51.86	5.92	26.57
5	13, 14, 15	2	2	3	1	47.23	47.20	50.42	48.28	3.42	28.34
6	16, 17, 18	2	3	1	2	49.05	50.53	51.64	50.41	1.69	31.77
7	19, 20, 21	3	1	3	2	55.46	57.32	55.79	56.19	0.98	35.06
8	22, 23, 24	3	2	1	3	50.48	58.02	54.94	54.76	4.37	22.31
9	25, 26, 27	3	3	2	1	50.61	52.75	50.91	51.42	1.34	32.94

**Table 6 sensors-23-09833-t006:** Taguchi ranking.

Experiment N°	Extrusion Temperature	Printing Speed	Bed Temperature	Layer Thickness
1	32.89	32.68	30.45	32.57
2	28.90	28.63	31.30	33.74
3	30.38	30.86	30.42	25.87
Δ	4.00	4.06	0.88	7.88
**Rank**	3	2	4	1

**Table 7 sensors-23-09833-t007:** UTS results for full factorial specimens.

Experiment N°	Specimen N°	LT	PS	ET	BT	Sample 1	Sample 2	Sample 3	Avg.	Avg. Comp
		mm	mm/s	°C	°C	Ultimate Tensile Strength MPa			MPa	MPa
7	19, 20, 21	0.6	30	205	80	52.46	56.33	55.79	54.86	54.86
8	22, 23, 24	0.6	40	205	80	52.63	53.72	54.36	53.57	53.57
4	10, 11, 12	0.4	30	205	80	55.63	55.86	57.33	56.28	52.64
5	13, 14, 15	0.4	40	205	80	56.47	56.40	54.70	55.86	52.13
9	25, 26, 27	0.6	50	205	80	50.45	52.39	52.91	51.92	51.92
1	1, 2, 3	0.2	30	205	80	51.62	54.23	49.80	51.88	51.88
6	16, 17, 18	0.4	50	205	80	55.47	56.41	54.50	55.46	51.77
2	4, 5, 6	0.2	40	205	80	52.19	52.35	50.67	51.73	51.73
3	7, 8, 9	0.2	50	205	80	52.23	50.90	49.03	50.72	50.72

## Data Availability

Data are contained within the article.

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
