# Peer review of "Design of Experiments to Compare the Mechanical Properties of Polylactic Acid Using Material Extrusion Three-Dimensional-Printing Thermal Parameters Based on a Cyber–Physical Production System"

_sensors, 2023, doi:10.3390/s23249833_

Round 1

Reviewer 1 Report

Comments and Suggestions for Authors

The paper significantly contributes to a better understanding of the design of experiments to compare PLA mechanical properties with material extrusion 3D printing thermal parameters based on a cyber-physical production system. The paper is well-written, and the overall methodology selected in this work is sound and based on best practices. The results are physically sound without any significant flaws being identified. However, minor suggestions are required for further improvement of this work:

Please cite and discuss recent publications that worked on the same study, and then the objective of this study should be compared with the literature. This would give the readers a clear idea of how this work differs from the previous study. Hence, this would help highlight the novelty.

Due to the increasing interest in different industries, the scientific community devoted remarkable efforts to these applications (3D Printing). Therefore, referring to and including the recent paper in the literature is worth mentioning:  https://doi.org/10.3390/app13020904

Please check the format for the in-text citations.

Section 2 should be replaced with Materials and Methods/Experimental Procedure, etc.

Figure quality should be improved. The labelled text should be clear.

Thanks.

Comments on the Quality of English Language

Minor editing of English language required.

Author Response

Dear Editors, 

The authors are pleased to resubmit for publication the revised version of sensors-2695895, entitled “Design of Experiments to compare PLA mechanical properties with material extrusion 3D printing thermal parameters based on a cyber-physical production system.”. The authors appreciate the comments provided by the 4 reviewers. The authors extend their appreciation for taking the time and effort necessary to provide such insightful guidance.

For the Reviewer 1 there is the following comments and responses:

Comment 1:  

"The paper significantly contributes to a better understanding of the design of experiments to compare PLA mechanical properties with material extrusion 3D printing thermal parameters based on a cyber-physical production system. The paper is well-written, and the overall methodology selected in this work is sound and based on best practices. The results are physically sound without any significant flaws being identified. However, minor suggestions are required for further improvement of this work:"

Response to Comment 1: 

The authors would like to thank the reviewer for his/her time in reading through our paper. The authors appreciate the comments and feedback provided to us.

Comment 2:

“Please cite and discuss recent publications that worked on the same study, and then the objective of this study should be compared with the literature. This would give the readers a clear idea of how this work differs from the previous study. Hence, this would help highlight the novelty.”

Response to Comment 2:

The main foundations of this research publications have it roots in publications on the same study, in fact in the second paragraph of the introduction section, there is a dedicated section where papers with similar approaches are referenced. The authors recognize that perhaps a more detailed explanation of them is necessary to give readers a clearer idea and highlight the novelty of this publication. As a note, one of the papers cited comes from the authors where a paper review on smart manufacturing technologies applied to the material extrusion 3d printing process served as the main foundation of the research carried out.   

Actions taken to comment 2:

 In paragraph 2 from the introduction section a detailed explanation of the paper main objective is explained to highlight its novelty. It is also presented the interconnection between material properties influence based on 3D printing process parameters and the need of a cyber physical system with experimental material models that are carried out based on the thermal properties of the process.

Comment 3:

“Due to the increasing interest in different industries, the scientific community devoted remarkable efforts to these applications (3D Printing). Therefore, referring to and including the recent paper in the literature is worth mentioning:  https://doi.org/10.3390/app13020904”

Response to Comment 3:

The authors thank the reviewer for providing this manuscript. Here, the authors would like to inform the reviewer that the main scope of the paper is to generate experimental material models based on 3D printing thermal parameters and static mechanical properties, we consider that for future research the fatigue consideration should be addressed. Additionally, we found that similar to the papers that were analyzed it was found that the material extrusion temperature has an impact on the final mechanical properties as well as the degradation of the material, therefore the paper mentioned was cited to reinforce the importance of the material extrusion temperature parameter in the process.

Comment 4:

“Please check the format for the in-text citations”.

Response to Comment 4:

The authors have reviewed the in-text citation format following the journal’s citation format.

Comment 5:

“Section 2 should be replaced with Materials and Methods/Experimental Procedure, etc.”

Response to Comment 5:

The author would like to thank the reviewer for this suggestion but still we consider that the main highlight of the publication is the application of the cyber physical system designed to obtain the experimental material model, therefore the second section is titled “Cyber Physical Production System and design of experiments.” The authors considers that initially a brief explanation of the cyber physical system is given and based on its characteristics then define the materials, method and experimental procedure employed.

Comment 6:

“Figure quality should be improved. The labelled text should be clear.”

The authors have reviewed the images quality formats and modified to their best way.

Reviewer 2 Report

Comments and Suggestions for Authors

Line 52 to 60, please add more reference beside your own literature and also on other polymers beside PLA

In the introduction the physical description about layer bonding is missing, please add the fundamentals and some modeling techniques like intimate contact, autohesive...

please specify the PLA grade in more detail

please summerise the stress strin curves in one plot

please also list failure strain

please explain why infill was used with 20%, i do not understand why solid test samples have been used

Author Response

Dear Editors, 

The authors are pleased to resubmit for publication the revised version of sensors-2695895, entitled “Design of Experiments to compare PLA mechanical properties with material extrusion 3D printing thermal parameters based on a cyber-physical production system.”. The authors appreciate the comments provided by the 4 reviewers. The authors extend their appreciation for taking the time and effort necessary to provide such insightful guidance.

For the Reviewer 2 there is the following comments and responses:

Comment 1:  

" Line 52 to 60, please add more reference beside your own literature and also on other polymers beside PLA"

Response to Comment 1:

The authors thank the reviewer for this meaningful observation and want to state that this publication have its main foundations on a paper review called “Smart Manufacturing Technologies Applied to the Material Extrusion 3D Printing Process” made form the authors, this publication contains more than 100 references in terms of mechanical material properties relations to 3D printing parameters, thermal processing conditions and sensor arrays employed to characterize and control this process. The authors recognize that perhaps a more detailed explanation of this fact should be addressed in the paper to give the reader more evidence support from the research carried out.

Actions taken to comment 1:

In paragraph 2 it is stated that the presented interconnection between material properties influence based on 3D printing process parameters and the need of a cyber physical system with experimental material models comes from the main review paper “Smart Manufacturing Technologies Applied to the Material Extrusion 3D Printing Process” to give more evidence supported on references

Comment 2:

“In the introduction the physical description about layer bonding is missing, please add the fundamentals and some modeling techniques like intimate contact, autohesive....”

Response to Comment 2:

The authors appreciated the feedback received, we consider that it is important to describe in more detail the physics and modeling involved during the adhesion of layers therefore it was a new paragraph dedicated to describing the 5 steps of the layer interactions during the process and why is important to monitor the thermal processes to acquire the mechanical properties required.

Comment 3:

“please specify the PLA grade in more detail”

Response to Comment 3:

The authors appreciated the observation that missing data from the material employed was not given therefore the main specification and grade type of the PLA employed is written.

Comment 4:

“Please summarise the stress strain curves in one plot”.

Response to Comment 4:

The authors would like to inform the reviewer that the graphs during the Taguchi analysis cannot be merged because there is no base for merging them. Each graph denotes a separate experiment, and no two graphs can be classified. As for the case for the full factorial data the Stress-Strain curves are merged in one plot.

Comment 5:

“Please also list failure strain.”

Response to Comment 5:

The author would like to thank the reviewer for this suggestion and on the table 3 the failure strain is included

Comment 6:

“Please explain why infill was used with 20%, I do not understand why solid test samples have been used”

The authors thank the reviewer for this comment. For the case of this analysis, the correlation between thermal properties with processing parameters to monitor and create an experimental material model does consider part design parameters as fixed and default variables from the Ultimaker Cura process simulator software. Such as Infill pattern, Build orientation, Layer Width and Infill Density. For the case of the last one, it was decided to use a percentage that would give reasonable rigidity to the sample but at the same time does not account for longer printing times due to the numerous experimental samples to test. Therefore, that is the reason why a 20% infill density was an optimal value for the design of experiments.

Reviewer 3 Report

Comments and Suggestions for Authors

  1. The paper does acknowledge various parameters affecting mechanical properties, but there's an overt emphasis on temperature. Given the multifaceted nature of 3D printing, other parameters might warrant exploration.
  2. A more detailed depiction of the CPPS setup and its implementation in the experiments would be advantageous, enabling other researchers to replicate or adapt the methodology.
  3. The control of the experimental environment might raise concerns. For instance, the influence of lab door operations on ambient temperature is noted, but the extent of this interference hasn't been explicitly elaborated upon.
  4. A more exhaustive literature review is recommended to position this research in context and delineate its relationship with existing knowledge. Specifically: (1) A comparison of 3D printing technologies used in other studies versus those employed here; (2) An evaluation of printing parameters selected in other researches vis-à-vis those chosen in this study; (3) Given the pronounced focus on thermal behavior during the printing process, further exploration into researches on 3D printing temperature management, thermal behavior, or mechanical properties related to temperature would offer a deeper understanding of the current state of the field.
  5. Consider incorporating on-site images or schematic diagrams to aid readers in grasping the experimental setup and the operational principles of CPPS. Such additions could be appended to supplementary materials.
  6. The discussion section could delve deeper into the relationship between individual parameters and mechanical properties, elucidating why certain relationships emerge. For instance, the correlation between layer thickness, printing speed, or printing temperature and mechanical properties could be further elaborated upon.
  7. For non-linear relationships or findings that might appear counterintuitive emerging from the results, a richer background or theoretical exposition would be beneficial.
Comments on the Quality of English Language

"The results showed that the order of the parameters in decreasing important order is layer thickness, printing speed, printing temperature, and bed temperature." A clearer expression is "The results indicated that, in descending order of importance, the parameters are: layer thickness, printing speed, printing temperature, and bed temperature."

Author Response

Dear Editors, 

The authors are pleased to resubmit for publication the revised version of sensors-2695895, entitled “Design of Experiments to compare PLA mechanical properties with material extrusion 3D printing thermal parameters based on a cyber-physical production system.”. The authors appreciate the comments provided by the 4 reviewers. The authors extend their appreciation for taking the time and effort necessary to provide such insightful guidance.

For the Reviewer 3 there is the following comments and responses:

Comment 1:  

" The paper does acknowledge various parameters affecting mechanical properties, but there's an overt emphasis on temperature. Given the multifaceted nature of 3D printing, other parameters might warrant exploration"

Response to Comment 1:

The authors thank the reviewer for this meaningful observation and want to state that this publication aims to construct a CPP system in terms of experimental materials models based on conditions monitoring of 3D printing processing parameters. Therefore, design parameters related to geometrical configurations are fixed in the whole process and thermal conditions can change due to material discontinuities or printer components performance. We acknowledge that variations from design parameters influence the final mechanical properties but in the case of this publication the CPP system is designed to monitor conditions, variations, and relations to mechanical properties, while design parameters remain fixed.

Comment 2:

“A more detailed depiction of the CPPS setup and its implementation in the experiments would be advantageous, enabling other researchers to replicate or adapt the methodology.”

Response to Comment 2:

The authors appreciated the feedback received, we consider that it is important to describe in more detail the CPPS designed, therefore the authors include a diagram explaining the design of the system.

Comment 3:

“The control of the experimental environment might raise concerns. For instance, the influence of lab door operations on ambient temperature is noted, but the extent of this interference hasn't been explicitly elaborated upon.”

Response to Comment 3:

The authors appreciated the observation derived from the environmental conditions generated during the experiments, there is a more insightful explanation of the consequences of this factor and addressed in the discussion section.

Comment 4:

“A more exhaustive literature review is recommended to position this research in context and delineate its relationship with existing knowledge. Specifically: (1) A comparison of 3D printing technologies used in other studies versus those employed here; (2) An evaluation of printing parameters selected in other researches vis-à-vis those chosen in this study; (3) Given the pronounced focus on thermal behavior during the printing process, further exploration into researches on 3D printing temperature management, thermal behavior, or mechanical properties related to temperature would offer a deeper understanding of the current state of the field.”.

Response to Comment 4:

The authors thank the reviewer for this meaningful observation and want to state that this publication have its main foundations on a paper review called “Smart Manufacturing Technologies Applied to the Material Extrusion 3D Printing Process” made form the same authors, this publication contains more than 100 references in terms of mechanical material properties relations to 3D printing parameters, thermal processing conditions and sensor arrays employed to characterize and control this process. Additionally, the authors submitted a second publication entirely devoted to the CPP system design. Due to the fact that explaining the whole details of it would be out of the scope of the main publication purpose. The authors recognize that perhaps a more detailed explanation of this fact should be addressed in the paper to give the reader more evidence support from the research carried out. Therefore, the authors state this fact in the introduction section. This publication is a third paper derived from whole research devoted to applications of smart manufacturing technologies applied to the material extrusion 3D printing process, where from literature review to CPP system design to experimental material models are developed to create a holistic approach to then being able to be controlled and auto regulated (future work).

Comment 5:

“Consider incorporating on-site images or schematic diagrams to aid readers in grasping the experimental setup and the operational principles of CPPS. Such additions could be appended to supplementary materials.”

Response to Comment 5:

The author would like to thank the reviewer for this suggestion and a new graph depicting the schematics of the methodology employed is presented.

Comment 6:

“The discussion section could delve deeper into the relationship between individual parameters and mechanical properties, elucidating why certain relationships emerge. For instance, the correlation between layer thickness, printing speed, or printing temperature and mechanical properties could be further elaborated upon.”

Response to Comment 6:

The authors thank the reviewer for this comment. A more elaborated discussion considering the correlations from the variables analysed is presented.

Comment 7

“For non-linear relationships or findings that might appear counterintuitive emerging from the results, a richer background or theoretical exposition would be beneficial”

 Response to Comment 7:

The authors appreciated the feedback received, we consider that it is important to elaborate more on the findings generated and in the discussion section this comment has been addressed.

Reviewer 4 Report

Comments and Suggestions for Authors

At first glance, the abstract is too long and lacks quantitative results. In addition, the purpose of the research has been clearly forgotten and the research achievements are not mentioned at the end of the abstract. So, the abstract is incompletely written and needs minor revision. The steps of the article should be mentioned first, and then the results should be presented quantitatively and qualitatively. Finally, the most important achievements should be mentioned.

The manuscript needs general writing and grammar editing. The novelty and purpose of the research should be clearly stated in the abstract and introduction.

The way of referencing in the introduction should be modified. The use of general sentences with more than four references can be seen in the introduction (Lines 30, and 31).

The introduction is brief. Also, the first paragraphs presented are primarily general and general information. At the end of the introduction, a suitable summary of the importance of the present issue should be provided. Also, discontinuity between paragraphs is evident in most of the introduction. It is suggested to rewrite the introduction.

Use the following resources to deepen the introduction and discussion. Shape memory performance assessment of FDM 3D printed PLA-TPU composites by Box-Behnken response surface methodology. 4D printing of PLA-TPU blends: effect of PLA concentration, loading mode, and programming temperature on the shape memory effect. 4D printing of PET-G via FDM including tailormade excess third shape.

What model is DOE based on? How is the selection range of printing parameters selected? How is the print quality checked? Explain more about the printing process. How has the reproducibility of these results been checked?

Considering the variable and different printing parameters, on what basis is the comparison of the printing ability of two materials done? It is better to introduce the optimal printing parameters at the end and compare the optimal results.

Some Figures of the article should be removed. For example, Figure 4 should be deleted and mention the standard of the tensile test is sufficient. Figures 1-3 can be merged or at least two figures can be deleted.

The results of mechanical properties should be presented in the same form so that the results can be compared. Also, convert the strain unit to percentage.

Comments on the Quality of English Language

***

Author Response

Dear Editors, 

The authors are pleased to resubmit for publication the revised version of sensors-2695895, entitled “Design of Experiments to compare PLA mechanical properties with material extrusion 3D printing thermal parameters based on a cyber-physical production system.”. The authors appreciate the comments provided by the 4 reviewers. The authors extend their appreciation for taking the time and effort necessary to provide such insightful guidance.

For the Reviewer 4 there is the following comments and responses:

Comment 1:  

" At first glance, the abstract is too long and lacks quantitative results. In addition, the purpose of the research has been clearly forgotten and the research achievements are not mentioned at the end of the abstract. So, the abstract is incompletely written and needs minor revision. The steps of the article should be mentioned first, and then the results should be presented quantitatively and qualitatively. Finally, the most important achievements should be mentioned."

Response to Comment 1:

The authors thank the reviewer for this meaningful observation a deeper review of the abstract has been done and comments have been included.

Comment 2:

The manuscript needs general writing and grammar editing. The novelty and purpose of the research should be clearly stated in the abstract and introduction.

Response to Comment 2:

The authors appreciated the feedback received, we consider that it is important to state the novelty of the research and it has been clearly stated in the abstract and introduction part.

Comment 3:

The way of referencing in the introduction should be modified. The use of general sentences with more than four references can be seen in the introduction (Lines 30, and 31).

Response to Comment 3:

The authors appreciated the observation from the referring format and this comment have addressed.

Comment 4:

“the introduction is brief. Also, the first paragraphs presented are primarily general and general information. At the end of the introduction, a suitable summary of the importance of the present issue should be provided. Also, discontinuity between paragraphs is evident in most of the introduction. It is suggested to rewrite the introduction”.

Response to Comment 4:

The authors thank the reviewer for this meaningful observation and want to state that in the introduction section a note referring to the fundamentals of this research has been included, this publication have its main foundations on a paper review called “Smart Manufacturing Technologies Applied to the Material Extrusion 3D Printing Process” made form the same authors, this publication contains more than 100 references in terms of mechanical material properties relations to 3D printing parameters, thermal processing conditions and sensor arrays employed to characterize and control this process. The authors recognize that perhaps a more detailed explanation of this fact is necessary to give readers a clearer idea and highlight the novelty of this publication and it is included.

Comment 5:

Use the following resources to deepen the introduction and discussion. Shape memory performance assessment of FDM 3D printed PLA-TPU composites by Box-Behnken response surface methodology. 4D printing of PLA-TPU blends: effect of PLA concentration, loading mode, and programming temperature on the shape memory effect. 4D printing of PET-G via FDM including tailormade excess third shape.

Response to Comment 5:

With due respect, the authors would like to inform the reviewer that after analyzing the publications suggested to reinforce the introduction or discussion section, it is still considered that those publications are out of the scope of the main objectives for the research carried out. As mentioned in the introduction section there is a complete paper review where the authors compiled relations with 3D printing processing parameters with mechanical properties and based on that the selection of the thermal parameters was determined. Furthermore, the objective of the research is to explore the solution of a continuous monitoring system for the thermal parameters where design parameters such as infill density remain fixed. It is appreciated the publications suggested but none of the main topics of them have a relation with the main topics on the publication developed, specially as the memory shape effect by 3d printing parameters is out of the scope.

Comment 6:

What model is DOE based on? How is the selection range of printing parameters selected? How is the print quality checked? Explain more about the printing process. How has the reproducibility of these results been checked?

Response to Comment 6:

The authors thank the reviewer for this comment. The authors want to clarify that the question concerned to the comment can be answered by the 3rd and 4th paragraph in the Design of experiment sections. As well as the reproducibility of the test by developing 3 samples per test and verifying the average value of them. Additionally, the thermal monitoring of  the printing conditions observe the same path for each samples reproduced 3 times.   

Comment 7

Considering the variable and different printing parameters, on what basis is the comparison of the printing ability of two materials done? It is better to introduce the optimal printing parameters at the end and compare the optimal results.

 Response to Comment 7:

The author would like to thank the reviewer for this meaningful observation, as for the main purpose of the research carried out is to stablish a potential tool to control and monitor processing conditions and mechanical properties of printed parts, the main objective for this paper was to determine if the thermal material model experiment could serve as a methodology to evaluate conditions monitoring and observe discrepancies on variables levels. To the date authors selected the most common material for this approach based on PLA and we recognize that for a different material the same approach and tool implementation can be used. 

Comment 8

Some Figures of the article should be removed. For example, Figure 4 should be deleted and mention the standard of the tensile test is sufficient. Figures 1-3 can be merged or at least two figures can be deleted”

Response to Comment 8:

The authors appreciated the observation regarding the figures mentioned. For the paper’s authors consider that the 3D printed specimen is a helpful graphic in order to let the readers what were the dimensions regarding the samples tested and possible replication or extent on the future research work. Additionally, we consider that merging Figures 1,2,3 would make them to lose scope and possibly the graphical potential to observe the main components of the CPP system designed.

Round 2

Reviewer 3 Report

Comments and Suggestions for Authors

The revised manuscript can be accepted in current form.